# A Unique Isolation of a Lytic Bacteriophage Infected *Bacillus anthracis* Isolate from Pafuri, South Africa

**DOI:** 10.3390/microorganisms8060932

**Published:** 2020-06-20

**Authors:** Ayesha Hassim, Kgaugelo Edward Lekota, David Schalk van Dyk, Edgar Henry Dekker, Henriette van Heerden

**Affiliations:** 1Department of Veterinary Tropical diseases, University of Pretoria, Faculty of Veterinary Science, Pretoria 0110, South Africa; lekotae@gmail.com (K.E.L); Henriette.VanHeerden@up.ac.za (H.v.H.); 2Unit for Environmental Sciences and Management: Microbiology, North-West University, Potchefstroom Campus, Private Bag X6001, Potchefstroom 2520, South Africa; 3Department of Agriculture Fisheries and Forestry, Office of the State Veterinarian, Skukuza 1350, South Africa; SchalkVD@nda.agric.za (D.S.v.D.); Atd@nda.agric.za (E.H.D.)

**Keywords:** bacteriophage infection, *Myoviridae*, anthrax, *Bacillus anthracis*, soil ecology, pathogen-bioremediation

## Abstract

*Bacillus anthracis* is a soil-borne, Gram-positive endospore-forming bacterium and the causative agent of anthrax. It is enzootic in Pafuri, Kruger National Park in South Africa. The bacterium is amplified in a wild ungulate host, which then becomes a source of infection to the next host upon its death. The exact mechanisms involving the onset (index case) and termination of an outbreak are poorly understood, in part due to a paucity of information about the soil-based component of the bacterium’s lifecycle. In this study, we present the unique isolation of a dsDNA bacteriophage from a wildebeest carcass site suspected of having succumbed to anthrax. The aggressively lytic bacteriophage hampered the initial isolation of *B. anthracis* from samples collected at the carcass site. Classic bacteriologic methods were used to test the isolated phage on *B. anthracis* under different conditions to simulate deteriorating carcass conditions. Whole genome sequencing was employed to determine the relationship between the bacterium isolated on site and the bacteriophage-dubbed *Bacillus* phage Crookii. The 154,012 bp phage belongs to *Myoviridae* and groups closely with another African anthrax carcass-associated *Bacillus* phage WPh. *Bacillus* phage Crookii was lytic against *B. cereus sensu lato* group members but demonstrated a greater affinity for encapsulated *B. anthracis* at lower concentrations (<1 × 10^8^ pfu) of bacteriophage. The unusual isolation of this bacteriophage demonstrates the phage’s role in decreasing the inoculum in the environment and impact on the life cycle of *B. anthracis* at a carcass site.

## 1. Introduction

*Bacillus anthracis* is a soil-borne, Gram-positive endospore-forming bacterium and the causative agent of the disease known as anthrax [1,2]. Anthrax is endemic in Kruger National Park (KNP), South Africa where outbreaks are cyclical among wildlife [3,4,5]. Anthrax is a zoonosis, but predominantly affects wild ungulates in the KNP [6,7]. *Bacillus anthracis* has a monomorphic genome that is closely related to other members of the *B. cereus sensu lato* group, specifically *B. cereus* and *B. thuringiensis.* Most of the genetic variation of these organisms occurs on their respective plasmids and pathogenicity islands which code for the toxins [8,9,10]. The plasmids of *B. anthracis* are pX01, which encodes for the lethal factor and edema factor [11,12], and pX02 that encodes for encapsulation [13]. There are also four putative lambdoid prophages that set *B. anthracis* apart from its close relatives [14]. 

While the transmission, dissemination, and conditions of persistence for *B. anthracis* during anthrax outbreaks have been well described over the past few decades [2,4,15,16], little is known about the bacterium’s activity in the soil during outbreak dormancy [17,18]. There have been several theories that are variations on Van Ness’ “incubator area” hypothesis [19] over the decades [2]. It is believed that the bacteria are capable of replicating within biofilms in the soil and rhizospheres of plants [17,20]. Schuch and Fischetti [18] proposed an alternate cycle of the bacterium in the biofilms of earthworm guts, mediated by bacteriophages and serving as replication sites. There is however a dearth of such environmental data from active anthrax outbreaks. Many of these theories focus on bacterial replication in the environment with less attention on the marked concentration decay of the inoculum over time.

Bacteriophages are viruses that infect bacteria. These viruses belong to the order *Caudovirales* with members of the *Siphoviridae, Podoviridae, Myoviridae*, and *Tectiviridae* families [21,22,23]. They have two pathways to replication within the bacterium: lysogeny and lysis. Lysogeny involves viral integration into the host genome, while lysis activates the host machinery to propagate its progeny. Ultimately, both pathways lead to bacterial lysis [23,24,25]. The difference between the two phage types is marked by the phage-encoded influences on the bacterium. Lysogenic bacteriophages have been theorized to impact bacterial lifecycles through fitness traits such as antibiotic resistance, replication efficiency, and sporulation rates of bacteria [18,26]. Bacteriophages have been useful tools for decades as a medium for diagnostics in microbiology, vectors in molecular biology, as well as in phage therapy applications [27,28]. The γ-phage is one such example used to identify *B. anthracis* in routine diagnostics [24,29]. The more popular focus for study in recent years has been in the evaluation of lytic bacteriophage endolysins as an alternative means of disinfection as well as therapy mediums in an age of dwindling antibiotic options [28,30,31]. In this study, we explore the role of a lytic bacteriophage in outbreak dynamics, as naturally occurring pathogen disinfectant agents while the phage propagates, using the example of a wildebeest (*Connochaetes taurinus*) carcass site during an anthrax outbreak in South Africa.

## 2. Materials and Methods 

The carcass remnants of a juvenile wildebeest (carcass # DS201579) was discovered on the 24th of February 2015 in Crooks Corner, Pafuri, KNP, South Africa. Approximately 75% of the carcass had already been consumed by vultures. White backed vultures (*Gyps africanus*) fed in large numbers and disrupt the carcass site a great deal during feeding. Generally, there is soil all over the carcass remnants. Blood smears, a swab (from the orbital socket) and soil samples (from the clearing where extravasation fluid marked the ground) were collected from the carcass site for diagnostics at the Skukuza State Veterinary Research Laboratory (SSRL) as part of the Skukuza State Veterinary Services Disease Surveillance System.

### 2.1. Isolation

The blood smear was Giemsa stained [2,32] and evaluated microscopically. The smear showed low level evidence of spores and the possible vestiges of encapsulated square ended *B. anthracis* vegetative cells. Since an anthrax outbreak was underway in the area at the time, the swab and soil were submitted for bacteriologic diagnostics of anthrax using selective and non-selective media. The swab was (i) directly streaked onto 5% impala blood agar (BA) and polymyxin EDTA-thallous acetate agar (PET; PLET excluding lysozyme) and then (ii) heat treated in a 100 µL phosphate buffered saline (PBS) at 65 °C for 30 min, thereafter it was similarly streaked onto the blood and PET agars and incubated at 37 °C overnight. One gram of soil was added to 9 mL PBS, shaken for 3 h, heat treated at 65 °C for 30 min, serial diluted (to 1 × 10^−8^), and then similarly plated out onto PET and BA media in triplicate for each dilution [2]. 

After the incubation, only the heat-treated BA plate produced two colonies (<5 mm in diameter) while the 2 PET agar plates and 1 BA plate from the direct streaking still appeared sterile. The colonies were sub-cultured and demonstrated sensitivity to 10 µg penicillin discs (Oxoid, United Kingdom) and diagnostic γ-phage. The single colonies were sub-cultured five more times thereafter until an opaque grey-white colony morphology was obtained (without any observed turbidity between 5–8 h post incubation) and then stored in glycerol. Other *Bacillus* spp. isolated on the media were also stored and identified.

The 2 PET and 1 BA plates (with sterile appearance) were then each flushed with 500 µL PBS and filtered through a 0.2 µm cellulose acetate membrane (Lasec, South Africa). A BA plate was divided into two with *B. anthracis* 34F2 Sterne strain plated onto the one half and *B. anthracis* DS201579 strain plated out onto the other. Ten microliters of the filtrate was dropped into the center of each half of the lawn area and the plate tilted in the dilution streak method [27] then incubated at 37 °C overnight. 

The filtrate, dubbed “Crookii,” was evaluated for plaque titers, in plaque-forming units (pfu), using the double layer agar method without the use of antibiotics [27,33] with *B. anthracis* DS201579 strain as the host and a 10 fold dilution series of the filtrate with five replicates for each dilution. Ten microliters of bacteriophage filtrate at each of the 10-fold dilution series was tested on the bacterial lawns of *B. subtilis* ATCC^®^ 6633™ (used as standard laboratory control), *B. cereus* ATCC^®^ 33019™, *B. thuringiensis* (lab strain –Pafuri) and *B. mycoides* (lab strain -Pafuri).

### 2.2. Viral Precipitation

Viral precipitation was achieved by dissolving 2.3% sodium chloride and 7% weight/volume polyethylene glycol 6000 (PEG6000) (Merck Millipore) into the filtrate and refrigerated overnight at 4 °C. The viscous solution was then centrifuged at 3000× *g* for 60 min at 4 °C. The viral pellet was dissolved to a final volume of 1 mL in phage buffer containing 100 mM calcium chloride (CaCl_2_), magnesium sulphate (MgSO_4_•7H_2_O), 50 mM Tris-Cl (1 M, pH 7.5), and distilled water. Half the precipitate was submitted for transmission electron micrograph (TEM) while the other underwent DNA extraction.

### 2.3. Transmission Electron Micrograph

The TEM was performed at the Electron Microscope Unit of the University of Pretoria, Faculty of Veterinary Science using negative staining with 2% uranyl acetate. 

### 2.4. The Effect of Viral Propagation on B. anthracis DS201579

To determine the effect of the bacteriophage *Bacillus* phage Crookii on *B. anthracis* under different conditions (simulating carcass nutrient conditions), 1 mL of cattle blood was inoculated with 500 spores of *B. anthracis* Sterne and *B. anthracis* DS201579 and 20 µL of Crookii at 5 × 10^8^ pfu/mL were incubated under standard atmospheric conditions or with 8% carbon dioxide at 37 °C overnight in a shaking incubator. Replicates included 0.8% w/v of (NaHCO_3_) sodium bicarbonate. A third set of positive control blood tubes included *B. anthracis* Sterne and γ-phage at 5 × 10^8^ pfu/mL, under the same conditions, in parallel. Blood smears (five slides for each set) were made with 20 µL of blood at 8 h, then 12 h and at 24 h. These smears were immediately fixed in methanol, stained with Romanowsky-Giemsa for 30 min [32] and examined at 1000x magnification. A 100 microscopic fields across a 100 µL of blood for each set of conditions were visually appraised and directly counted to reflect cell proliferation. The blood smear analyses were aimed at determining if sporulation is triggered as a defense mechanism against phage infection [26,34] or in response to nutrient availability [35]. Since there are multiple variables to consider, figures illustrating the descriptive statistics of the vegetative cell and spore counts were generated in R Console version 3.2.1 [36]. 

The DNA from *B. anthracis* DS201579 strain was isolated using High Pure Template Preparation Kit^®^ (Roche) using the Gram-positive bacterial protocol with 20 mg/mL Lysozyme L6876 (Sigma Aldrich). The viral DNA was isolated from propagated lysates through the phenol-chloroform isoamyl alcohol method from phage lysate at a concentration of 9 × 10^11^ pfu/mL [37,38].

### 2.5. High-Throughput Sequencing

Shotgun library preparations of the *B. anthracis* and *Bacillus* phage Crookii were performed using the Nextera XT DNA Sample Prep Kit (Illumina, USA) by following the manufacturer’s instructions. Sequencing was performed on the Illumina MiSeq sequencing platform (Illumina) using the MiSeq Reagent Kit v3 (600 cycle).

#### 2.5.1. Bioinformatics Analysis

The quality of the sequenced 300 bp paired end reads of the *B. anthracis* DS201579 and *Bacillus* phage Crookii were assessed using FASTQC software version 0:10.1 [39]. Adaptors and ambiguous nucleotides sequences were trimmed using CLC Genomic Workbench 7.5.1. The reads were quality trimmed to an average length of 288 bp. Sequence read mapping analysis of the *B. anthracis* DS201579 reads was performed on CLC Genomic Workbench 7.5.1 using *B. anthracis* Ames Ancestor (NC_007530.2; NC_007322.2 and NC_007323,2) as a reference. Unmapped reads from the read-mapping analysis were collected and *de novo* assembled using CLC Genomic Workbench. The *de novo* assemblies of the *B. anthracis* DS201579 were carried out using CLC Genomic workbench 7.5.1. The assembled contigs of the bacterium were aligned with BLASTn [40] using *B. anthracis* Ames Ancestor (NC_007530.2; NC_007322.2 and NC_007323,2) as a reference genome. The progressive Mauve tool [41] was used to re-order the assembled contigs using *B. anthracis* Ames ancestor genome. Prophage regions were predicted using PHAge Search Tool Enhanced Release (PHASTER) [42].

*De novo* assembly of the *Bacillus* phage Crookii was also carried out using CLC Genomic Workbench 7.5.1. BLASTn was used to determine the closest sequence identity to *Bacillus* phage Crookii [40]. The CGView comparative tool [43] was used to compare the *Bacillus* phage Crookii genome sequence to *Bacillus* phage WPh. Phylogenetic analysis of the bacteriophage was performed to identify and classify *Bacillus* phage Crookii. The phage major capsid protein coding gene sequence (~1,430 bp) was extracted from the assembled genome of *Bacillus* phage Crookii. The sequence was used to mine for the other closely related phage sequences using BLAST homology searchers from NCBI. Multiple alignments of the extracted gene sequences were performed using MAFFT 7 [44]. Maximum likelihood analysis of the *Bacillus* phage Crookii and related *Bacillus* phage sequences were performed using 1000 bootstrap replications in MEGA 7 [45]. The assembled contigs of *B. anthracis* DS201579 and *Bacillus* phage Crookii were annotated using the NCBI Prokaryotic Genome Annotation Pipeline (PGAP) [46] and Rapid Annotation using Subsystem Technology (RAST) [47] respectively.

#### 2.5.2. Nucleotide Sequence Accession Numbers

The genomes of *B. anthracis* DS201579 and *Bacillus* phage Crookii have been deposited in GenBank under the accession numbers LVWF00000000 (includes complete pX02 accession CM008136) and KU847400 respectively.

## 3. Results

While the same protocols were used as per the standard operating procedures of the veterinary laboratory, SSRL; it is important to highlight that the first few attempts at isolation from the samples at this carcass site resulted in sterile media. This was true for both the selective media plates and the 5% BA plates after 24-h incubations. The successive attempts on the same samples (that yielded an isolate as described below) were pursued because of the observation of (i) “ghost cells” and spores on the Giemsa stained blood smears, (ii) the visual evidence from the carcass site and (iii) the proximity to other anthrax positive carcasses in the area during an active outbreak. 

### 3.1. Isolation

Two colonies (i.e., total of 2 CFU from the heat treated swab) on BA plate had a greyish-white, rough, domed morphology and were sensitive to penicillin and γ-phage. This preliminarily identified the bacterial isolates to be *B. anthracis* [2,48]. When a whole plate of a single colony was streaked out, it took on a turbid appearance after 8 h and thereafter developed large plaques (≥ 10 mm in diameter) in the lawn. 

The soil sample failed to produce any B. *anthracis* isolates at any of the dilutions, despite heat treatment. The plates were also conspicuous for the absence of *B. cereus* colonies which are normally abundant in soil samples from Pafuri. There was, however, a consistence of *B. subtilis* on all the plates with *B. mycoides* featuring on only the plates from the lower dilution series (≤1 × 10^-3^). *Bacillus* spp. were identified using classic bacteriologic methods at the Bacteriology Laboratory of the Department of Veterinary Tropical Diseases, Faculty of Veterinary Science, University of Pretoria.

The filtrate that was placed in the center of the *B. anthracis* Sterne 34F2 and DS201579 bacterial lawns demonstrated clear lytic zones at the deposit sites (Figure 1). The dilution streak at the edge of the plate demonstrated a smaller lysis zone for Sterne 34F2 than the DS201579 strain (Figure 1). The DS201579 strain had also taken on a turbid appearance and the lysis zone began to spread a further 10 mm in diameter for the ensuing 4 h. This indicated the presence of an infective agent such as a bacteriophage rather than the presence of an antimicrobial compound (which was also initially considered) in the filtrate. The viral filtrate concentration was determined to be 6.8 × 10^8^ pfu/mL by plaque enumeration. For this reason, the *B. anthracis* isolate DS201579 was sub-cultured until such time as single colonies remained white and opaque, after plating, without developing plaques after 6 h at room temperature. This was to ensure that the isolates were “purified” of any latent bacteriophages.

*Bacillus* phage Crookii counts were stable up to 50 °C for 15 min in Dulbecco’s PBS buffer, however, there was no survivability after 60 °C for 15 min. This is why at least the 2 CFU could be obtained from the heat treated swab; which allowed the spores to germinate and replicate. Subsequent serial passage (sub-culturing) of isolates appeared to “cure” the bacterium of the bacteriophage. The filtrate was also tested against bacterial lawns of *B. cereus*, *B. thuringiensis*, *B. subtilis*, and *B. mycoides* strains in order to determine its specificity. The phage was only lytic against *B. anthracis*, *B. cereus*, and *B. thuringiensis*, but demonstrated a greater affinity for encapsulated *B. anthracis* strains. *Bacillus cereus* only demonstrated plaques at very high viral concentrations (>10^9^ pfu), while *B. thuringiensis* revealed clear plaques only at the highest concentrations (>10^11^ pfu)

### 3.2. Transmission Electron Micrograph

The identification of the bacteriophage was based on viral morphology fitting within the criteria for *Myoviridae* viruses (Appendix A): icosahedral head 60–145 nm; elongated heads 80 × 110 nm and tail 16–20 × 80–455 nm according to Ackermann, (2011).

### 3.3. The Effect of Viral Propagation on B. anthracis DS201579

In order to determine the activity and effect of *Bacillus* phage Crookii on the bacterium at the carcass site, the virus was propagated under different conditions including sodium bicarbonate which is a natural trigger for encapsulation [35,49]. The mean spore counts were directly tabulated from 100 microscopic fields of Romanowsky-Giemsa stained smears taken at 8 h (to allow for spore germination and thereafter replication), 12 h, and 24 h (Appendix A). As indicated the blood smear analyses were to determine whether sporulation triggered a defense mechanism against phage infection or whether it is a response to nutrient availability. A graphical representation of the different conditions can be seen in Figure 2 where typical microscopic fields were visualized for cell enumeration. Appendix B includes a descriptive analysis of the interaction between the bacteriophage, host bacterium, and nutrient availability in the environment. The blood smears revealed that encapsulated *B. anthracis* strain DS201579 fared much better than unencapsulated Sterne strain in the changing blood conditions (Appendix A, Figure A1, Figure A2A–C). This could be seen at 12 h counts where all the Sterne strain cells had sporulated, while the DS201579 strain still had replicating vegetative cells. The presence of the bacteriophages also resulted in lower end point bacterial counts for both bacterial strains (Appendix A, Figure A3A–D). Sterne strain was similarly affected by both *Bacillus* phage Crookii and γ-phage. Despite strain DS201579 demonstrating a better replication efficiency, the presence of *Bacillus* phage Crookii resulted in lower end point counts than Sterne vaccine strain (Appendix A, Figure A4A–D). For both bacteria sporulation was triggered earlier during the logarithmic phase in the presence of bicarbonate, carbon dioxide, as well as the presence of a lytic bacteriophage (Appendix A, Figure A5A–E, Figure A6A–E, Figure A7A–F).

### 3.4. Genomic Features of Bacillus Anthracis DS201579

About 140X sequenced coverage of the trimmed reads were used to assemble the 32 contigs of *B. anthracis* DS201579 genome (Appendix A). The genome features of *B. anthracis* DS201579 presented a chromosome of 26 contigs, two plasmids pX01 of four contigs and pX02 assembled as one contig (Appendix A). The genome coverage of *B. anthracis* DS201579 (5.45Mb) was about 99% with respect to *B. anthracis* Ames ancestor (5.50 Mb). An average G+C content of 35% was determined in the *B. anthracis* DS201579. Annotation of the bacterial genome using NCBI Prokaryotic Genome Annotation Pipeline presented 5724 coding sequences (CDSs) with a total of 94 RNA genes. Five prophages were determined in the chromosome of *B. anthracis* DS201579. The prophages nucleotide sequences were not 100% identical with the query or given sequences in GenBank. This included the LambdaBa01, LambdaBa02, LambdaBa03, LambdaBa04 and a partial prophage referred to in general terms as Phage_Bacilli pfEDR_5_NC_031055 (Appendix A).

Read mapping analysis of *B. anthracis* DS201579 determined that it consists of chromosomes, pX01 and pX02. Analysis of the unmapped reads of the DS201579 strain from the read mapping analysis resulted in eight contigs with less than 400 bp size and low sequence coverage on the contigs (Appendix A). The eight contigs had a BLASTn homolog to partial/minor phages with no resemblance to either of the bacterial prophages nor *Bacillus* phage Crookii. They had similar sequence identities of ≤90% with *Bacillus* phage WPh. Neither of these identified contigs are similar when aligned to *Bacillus* phage Crookii. This is indicative of multiple possible phage infection events of *B. anthracis* strain DS201579 by *Myoviridae* environmental phages over time.

#### Genomic Features of *Bacillus* Phage Crookii

*De novo* assembly of the *Bacillus* phage Crookii resulted in a single contig with 70X sequence coverage. The bacteriophage identified as *Bacillus* phage Crookii is a dsDNA *Myoviridae* virus ~154,012 bp with a G+C content of 37%. The number of coding sequences identified by RAST for this phage was 235. About 94% of the annotated CDS’s are hypothetical while 6% have a known protein nomenclature. This included the lysins, head/capsid proteins, cell wall binding proteins, base plate proteins, replication, and packaging machinery (Appendix A). No intergrase/recombinase proteins were determined in the *Bacillus* phage Crookii, confirming it to be lytic in behavior. The genome properties of *Bacillus* phage Crookii codes for lysine *N*-acetyl-muramoyl-*L*-alanine amidase (endolysin), which is responsible for bacterial lysis. It contained multiple tRNA genes namely, tRNA-Asn-GTT (asparagine) and tRNA-Asp-GTC (aspartate). It also had coding regions for the metal-dependent hydrolases of the metallobeta-lactamase superfamily and RNA polymerase sigma factor SigB (Appendix A).

BLASTn comparison of *Bacillus* phage Crookii on NCBI revealed a viral strain relationship to a *Myoviridae Bacillus* phage WPh., isolated from an anthrax positive zebra carcass in Namibia [50,51]. The whole genome of *Bacillus* phage Crookii was compared and aligned with *Bacillus* phage WPh. (Appendix A). The genome size of *Bacillus* phage WPh. is 156,897 bp. *Bacillus* phage Crookii demonstrated 83% query cover and 85% sequence identity with WPh accession number HM144387.1. The G+C content of *Bacillus* phage WPh phage is 36.4%, which compares well with *Bacillus* phage Crookii. The same number of tRNA’s were observed on the genomes of both phages. Phylogenetic analysis of *Bacillus* phage Crookii showed that it groups with other related *B. cereus*-group phages (Figure 3). *Bacillus* phage Crookii also groups closely in the same cluster with *Bacillus* phage WPh.

## 4. Discussion

This study is the description of the unique isolation of a lytic bacteriophage, *Bacillus* phage Crookii infected *B. anthracis* DS201579 strain from a wildebeest carcass in Pafuri close to Crooks corner in KNP. In the diagnostic evaluation of the carcass site DS201579, the apparent “sterility” of the untreated blood agar plates drew special attention to the samples from this site. The novelty of the identification of *Bacillus* phage Crookii and its inherent appetite for *B. cereus, B. thuringiensis* and *B. anthracis* in culture is what sets this bacteriophage isolation apart. The heat-treated samples from the carcass site yielded *B. anthracis* spores, but only two small colonies on enrichment medium from a biological sample is unusual. When sub-cultured, plaques once again appeared over time, indicating that even these spores failed to escape infection. The inability to obtain isolates from the swab before the heat treatment indicates that the bacteriophage was active in the sample.

Bacteriophages are not able to infect bacterial spores [52], but they are able to harbor within spores after infection of a vegetative cell [53,54]. The bacteriophage infected bacteria are also able to persist in biofilms, created by bacteria, in the environment [18,55]. The tailed phages, of which *Myoviridae* phages form a part, tend to be the most resilient, abundant, and stable to environmental factors [22,56]. This does not mean that these phages will survive for prolonged periods outside of their ideal conditions. Phages are certainly sensitive to external environmental factors such as temperature and pH ranges [56,57]. This highlights why multiple bacterial hosts would be required to subsist between anthrax outbreaks in the environment. That said; *Bacillus* phage Crookii demonstrated a much greater affinity for encapsulated *B. anthracis* strains from the anthrax endemic region of Pafuri, however, it could also infect unencapsulated *B. anthracis* Sterne and *B. cereus,* but only at much higher viral concentrations. Under such conditions, it can be said that, at low concentrations of *Bacillus* phage Crookii there is a higher specificity for pathogenic *B. anthracis* in the anthrax enzootic region of Pafuri. The wider host range for other *B. cereus*-group members may just be due to a similarity in receptor sites and in the wider context as a phage survival mechanism.

To explore factors that influence bacteriophage infection of the bacterium, we simulated decomposing carcass conditions. The spores from un-encapsulated *B. anthracis* Sterne versus the encapsulated *B. anthracis* DS201579 strains were incubated under different conditions of oxygen concentration, pH and bicarbonate content as a control for comparison of the phage effects. The inclusion of 8% CO_2_/bicarbonate was to stimulate thickening of the pathogenic *B. anthracis* capsule [49,58] and to mimic carcass blood conditions [59]. The germination of the spores was lower in the presence of sodium bicarbonate, under standard incubation conditions, and even lower when combined with 8% CO_2._ This is expected because of the inhibitory nature of the combination of CO_2_ [60] and sodium bicarbonate [61] in the germination of soil-borne spore formers. Since bacteriophages cannot infect bacterial spores [52,54]; sporulation before infection by the phage would ensure the bacterium’s persistence in the environment. The blood smear analyses were aimed at visually appraising whether sporulation is triggered as a defense mechanism against phage infection [26,34] or in response to nutrient availability [35] through cell enumeration. From the results, sporulation is in response to nutrient availability as well as in response to the presence of a lytic bacteriophage. This demonstrates the complexity of factors which affect the survival and persistence of inoculum in the environment.

*Bacillus* phage Crookii proved more lytic than Gamma (γ)-phage to DS201579 based on plaque zones at the same bacteriophage concentration. There seemed to be no difference in the effect between Crookii and Gamma on the un-encapsulated Sterne strain. *Bacillus anthracis* DS201579 did not appear to develop any phage resistance characteristics over time, escaping lysis only through sporulation. Sporulation and replication activity were due to nutrient availability. The presence of bacteriophages also resulted in early onset sporulation. It has been suggested that this could be a bacterial survival mechanism against bacteriophage infection [34]. The presence of the sodium bicarbonate and the carbon dioxide inhibited spore germination irrespective of the presence of the phages and therefore resulted in higher end point spore counts due to the inability of the phages to infect the bacteria. The lower bacterial counts from samples which included bacteriophages demonstrates the aggressive ability of environmental phages like *Bacillus* phage Crookii to reduce overall bacterial counts at the endpoint. This is due to the fewer vegetative cells available for replication during the 0 to 12 h logarithmic incubation period and its coinciding with the highest infectivity period of the phage [25,62]. This suggests a more opportunistic infection ability of the bacteriophage.

The smears were also used to evaluate whether the bacteriophages would lyse all available vegetative cells under optimum conditions of bacterial replication. It appears as though the sporulated bacteria have escaped infection, but when these spores are plated out, phage plaques once again become apparent. It has been suggested that bacteriophages are able to “read” the quorum signals of their hosts and are therefore likely to halt further bacterial lysis at the peak of host death [63]. The bacteriophages can then remain dormant within the host until germination and replication is triggered. Since spores are far more stable and resilient to environmental conditions, this allows persistence of the bacteriophage within the soil environment and host population.

While the lytic effects of the bacteriophage on *B. anthracis* are quite apparent, a genetic analysis of the bacterium and virus highlighted an alternative relationship as well. The genome of *Bacillus* phage Crookii included a potential coding region for the metal-dependent hydrolases of the metallobeta-lactamase superfamily, which was previously described in *B. cereus* for conferring resistance to carbapenems [64,65] to prevent premature lysis of the bacterial cell by external factors [56]. No holin coding region was observed in this genome; however, a large number of hypothetical proteins in the genome were specified to still require annotation. Comparative sequence analysis identified a close relative to *Bacillus* phage Crookii in the form of *Bacillus* phage WPh. *Bacillus* phage WPh. was isolated from an anthrax positive zebra carcass in Etosha National Park, Namibia [50,51]. Both phages are *Myoviridae* dsDNA bacteriophages based on similar GC content and sequence identity with characteristics of *Spounaviruses* [51,66]. These bacteriophages infect members of the *B. cereus sensu lato* group. It can be seen that the relationship between these two bacteriophages, which are associated with anthrax carcass sites on opposite sides of the African continent, are much closer than those isolated from *B. cereus, B. thuringiensis* and other *Bacillus* spp (Figure 3).

*Bacillus anthracis* DS201579 genome features consists of the chromosomes, pX01 and pX02. The prophages determined in this study were also reported to be present in *B. anthracis* Ames ancestor [67]. Different sizes of the prophage regions were also observed in the genome of *B. anthracis* DS201579. PHASTER analysis of the bacterial genome revealed only the four prophage regions that have been well described for *B. anthracis* [9,14,68] as well as, a previously characterized partial prophage [14,69]. A separate partial phage was also identified in the analysis of the unmapped reads of the bacterial genome. This phage had similarities with *Bacillus* phage WPh., but the contigs did not align with *Bacillus* phage Crookii. This is an indication of past infection events of the bacterium. Phage infection by virions already present in the bacterium can confer phage resistance to infection by new bacteriophages as well as through origin-derived phage-encoded resistance [70,71], however, this was not the case for *B. anthracis* DS201579 and *Bacillus* phage Crookii.

## 5. Conclusions

*Bacillus* phage Crookii is an aggressively lytic dsDNA *Myoviridae* bacteriophage infecting *B. anthracis* in the environment of Pafuri, KNP. The phage presented itself as an unusual isolate during diagnostics from a carcass in an anthrax outbreak. Since no bacteria could be isolated from soil and only heat treatment greater than 60 °C yielded *B. anthracis* from a blood swab, it demonstrates the role of environmental bacteriophages in the natural reduction of bacterial inoculum at carcass sites.

Whole genome sequencing of *B. anthracis* DS201579 revealed the presence of the lambdaBA phages that have been previously described in Ames ancestor and commonly found among *B. anthracis*. No bacteriophages, other than the expected four (Lambda BA01, BA02, BA03, and BA04) and Phage_*Bacilli*_pfEDR_5, were identified within the host bacterial genome as determined by whole genome sequencing.

The remnants of another partial phage within the *B. anthracis* DS201579 strain is suggestive of multiple possible phage infection events of the *B. anthracis* strain (DS201579) by *Myoviridae* environmental phages over time. Despite this, *B. anthracis* DS201579 did not manifest phage resistance characteristics against *Bacillus* phage Crookii even after serial passage. The initial isolation of the bacterium highlighted the bacteriophages ability to harbor within spores, thus persisting alongside the bacterium until germination.

*Bacillus* phage Crookii demonstrated a viral strain sequence identity relationship to *Bacillus* phage WPh. The sequence similarity of these hypervariable environmental *Myoviridae* bacteriophages is descriptive of the common ecology within African soil biomes despite geographical disparity.

The identification of *Bacillus* phage Crookii was fortuitous but was indicative of how many phages must be missed during routine bacterial diagnostics. Recent studies have shown that the isolation methods popularly employed in the isolation of bacteriophages from water and soil biomes (including this study) select against larger, more complex bacteriophages [72]. *Bacillus* phage Crookii and other lytic bacteriophages represent an efficient pathogen bioremediation mechanism in the environment. They could prove to be the missing link in determining outbreak termination in endemic/enzootic regions. More study is required for understanding the complexity of soil-borne factors influencing anthrax outbreaks.

## Figures and Tables

**Figure 1 microorganisms-08-00932-f001:**
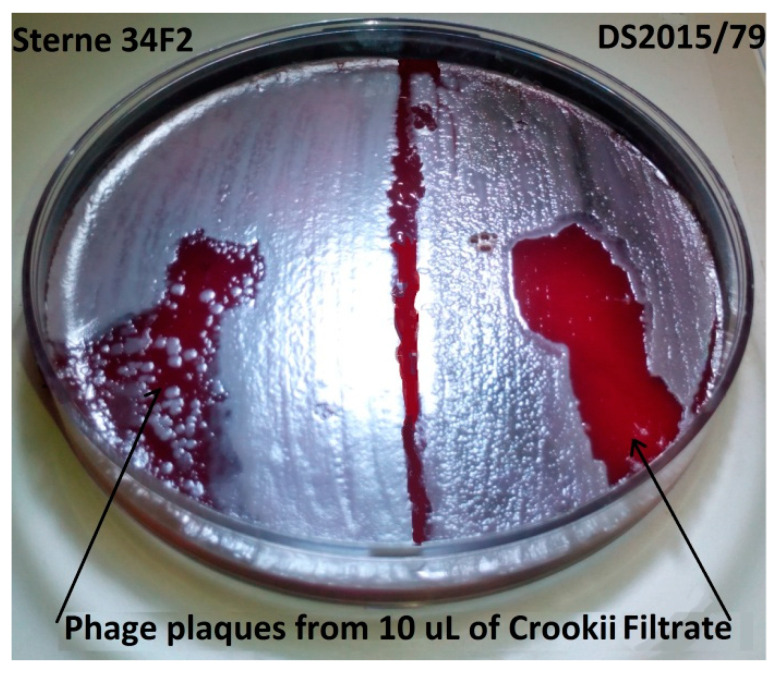
Blood agar plate with *Bacillus anthracis* bacterial lawns of Sterne 34F2 (**left**) and virulent DS201579 (**right**) after 12 h incubation at 37 °C. The plaques indicated by the arrows are where the bacteriophage *Bacillus* phage Crookii has lysed bacterial cells. The phage demonstrates a greater affinity for *B. anthracis* DS201579 than Sterne strain; where the DS201579 lawn is also displaying turbidity indicating phage infection.

**Figure 2 microorganisms-08-00932-f002:**
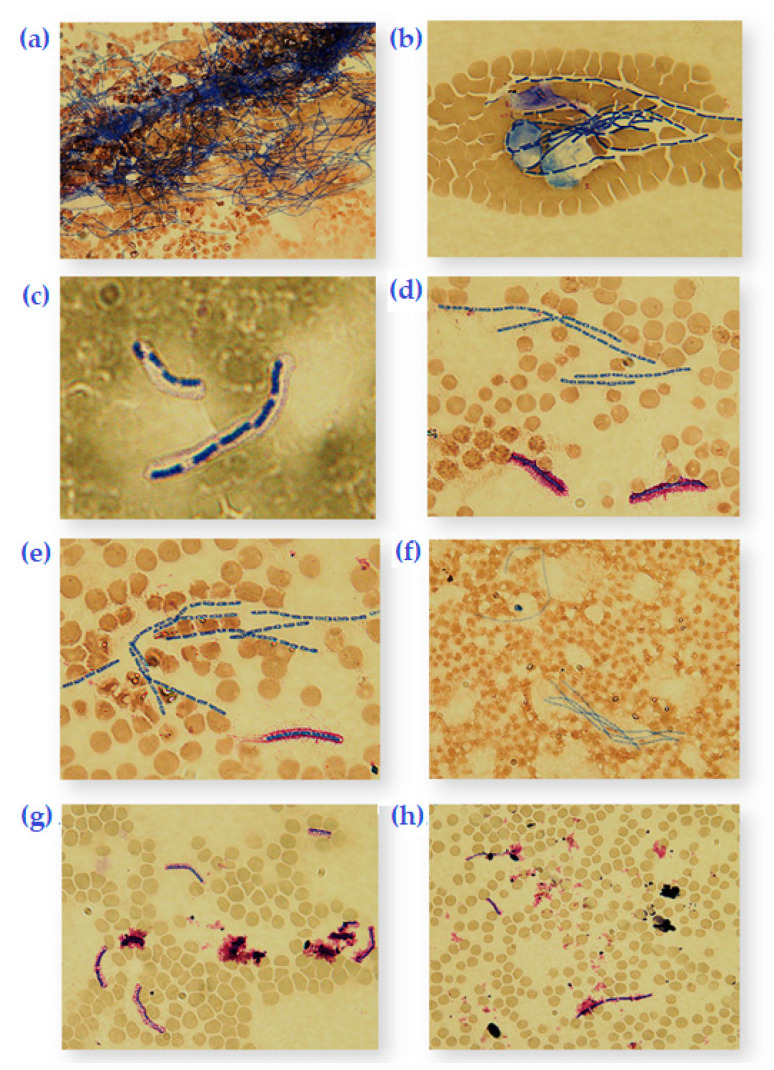
Romanowski-Giemsa-stained blood smears of *Bacillus anthracis* at 1000x magnification. (**a**) Unencapsulated *B. anthracis* Sterne vaccine strain at 12 h of 37 °C incubation. (**b**) Sterne with γ-phage at 8 h of 37 °C incubation. (**c**) *B. anthracis* DS201579 showing encapsulation (balloon-like layer) at 8 h of 37 °C incubation. (**d**) DS201579 with NaHCO_3_ in 8% CO_2_ at 8 h of 37 °C incubation showing thickened capsule (purple) and cells with endospores (blue). (**e**) DS201579 at 37 °C incubation for 12 h in 8% CO_2_ where all cells contain endospores (blue) and degradation of mother cell has begun (purple streaks). (**f**) DS201579 with NaHCO_3_ and 8% CO_2_ at 37 °C incubation for 12 h of showing 100% sporulation. (**g**) DS201579 with *Bacillus* phage Crookii at 37 °C incubation for 12 h and 8% CO_2_ of showing lysed cells as well as intact endospores. (**h**) DS201579 with *Bacillus* phage Crookii at 37 °C incubation for 12 h where all vegetative cells in view were lysed.

**Figure 3 microorganisms-08-00932-f003:**
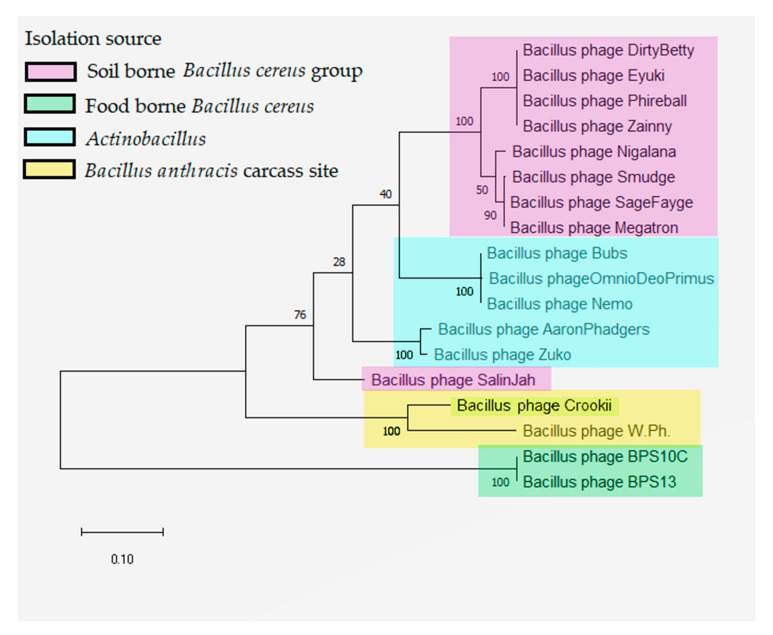
Maximum likelihood phylogeny for the *Bacillus* phages based on the phage major capsid protein showing the clustering of *Bacillus* phage Crookii (highlighted in yellow along with anthrax associated *Bacillus* phage WPh). Bootstrap values >28% are indicated at the internodes. Bacteriophages infecting *Bacillus* spp. cluster according to their isolation source (indicated in the color legend).

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
