# Peer review of "A Unique Isolation of a Lytic Bacteriophage Infected Bacillus anthracis Isolate from Pafuri, South Africa"

_microorganisms, 2020, doi:10.3390/microorganisms8060932_

Round 1

Reviewer 1 Report

A Unique isolation of a lytic Bacteriophage Infected Bacillus anthracis Isolate from Pafuri, South Africa by Hassim et al.

The manuscript is well written and understandable to the reader. It is easy to follow the thoughts and argumentation of the authors during their presentation of the results and their discussion. In general, the presented content is worthy for publication and shows interesting ideas for future applications. It provides, insights into the bacteriophages-world of Bacillus cereus-group. However, two things are to mention prior accepting this manuscript for publication, in order to improve this study:

  • The manuscript seems to be written in a rush. Lots of small errors or mistakes can be seen. Inconsistency of wording and formatting. It is a must, to review the text once more before publishing. More in detail later
  • The manuscript seems to be written by bacteriologist and not by molecular biologist and definitively not by experts in Next-Generation-Sequencing. Especially for the latter subject, strong weaknesses occur in the textbody, which is ok for oral discussions (‘lab-slang’), but which is scientifically not correct or insufficient described. I guess, that the work should be revised extensively, which really improves the impact of the study, but more in detail later. With writing more manuscripts, presenting and discussing NGS sequencing, it will become better.

Writing and Inconsistency:

  • Italize everywhere(!) Bacillus, Bacillus cereus, Bacillus anthracis etc. Good at the beginning, bad in the middle, acceptable at the end. Similar for virus-families (Myoviridae e.g.) Italize everywhere. Even in figure legends.
  • Check “comma” for “1,000”-markings. Do not know the English word of it. Some are missing. Some are present.
  • Line 82: “One gram of soil”
  • Line 90: Bacillus spp. isolated (not Isolated). Yes, MS Word has an auto-correction tool, but you have to check its suggestions.
  • Line 134: “Nextera”
  • Line 143: Check white space after DS201579
  • Italize “de novo” everywhere (line 149, e.g.)
  • Line 182: Check Spelling of Laboratory. I guess you mean “Laboratory” not “laboratory”
  • Line 219 “indicated”
  • Line 262 check white space before 90%
  • Line 320 Check if “.” Is subscribed or not (latter would be correct)

Scientific Remarks

  • Abstract line 19. Was this bacteriophage now environmental (soil-borne) or a veterinarian sample (wildebeest carcass). Correct the sentence.
  • Abstract line 26 “but more”. Just remember these two words. “more” means, that it can be counted or measured. There is no quantitative answer in this text to support this item!
  • Line 65: “naturally occurring pathogen disinfectant agents”. Nice wording for a talk, but not for a written paper. Humans can think about it, to use it as disinfectant, but the bacteriophage does not intend to be disinfectant. It intends to survive!!
  • Line 102: You tested B. cereus, 2x B. anthracis, B. thuringiensis, B. subtilis and B. mycoides. You did not test “B. anthracis var cereus” That is missing definitively. Order the species to cereus-group and not-cereus group, as you mentioned, that the bacteriophage is specific to B. cereus-group members. Did you really tested B. subitilis?
  • Line 126: Nice to know that you used Rstudio, but this is not important. You used “R” in version 3.5 or similar. Mention and cite this. You even used MS Word and did not write the manuscript with Windows. Rstudio is just an environment for using R in a more easy way.
  • Line 135: please have a look into the recommendation of the the Nextera Kit. I am pretty sure, that you did not use 0.2ng of template!! Check this
  • Line 136: “diluted in hybridization buffer” It looks, that you do not know what you did. The libraries are diluted and formamide was added, if you are using Illumina MiSeq. Including a phiX as run-control. There is not really a "hybridization" buffer. And this step belongs to “Preparing and loading of the Sequencer” not to the section “Preparing of library”. Just skip this sentence.
  • Line 143: de novo assembly was performed using software suite “CLC Genomic workbench”. But which assembler was used. Please add.
  • Line 143 “The assembly contigs”. Did you mean “the assembled contigs”? To be honest, I do not really know, what you did here. Did you blast the contigs in order to confirm, that it is B. anthracis? Mapping and re-ordering of contigs/scaffolds was mentioned in the next sentence
  • Line 146: did you really use “mauve” and not “progressive mauve”, which is implemented in mauve. Please check and correct.
  • Line 154: “multiple alignments…was performed”. Which alignments? Replicates? Several ORFs? Which ones. Please add information.
  • Line 157: “Bootstrap replications” not “interactions”
  • Line 167: Raw reads are missing. Provide raw reads (fastq-files) to the readers by submitting them to NCBI or ENA of both samples, phage and B. anthracis
  • Line 175: No, this did not confirm to be B. anthracis. It preliminary identified the sample as Ba , but did not confirm it to be. No molecular results (B. anthracis specific PCR) was presented. The confirmation was done with the NGS-sequencing. Rephrase and correct the sentence. Try to be scientifically correct.
  • Line 248: “about 140x sequenced reads”???? I know what you mean, but that is not, what you wrote. You had a coverage (sequencing depth) of 140x.
  • Line 248: 280-bp??? No, you had 2x 300bp paired end reads. The sequencer sequenced 300 and something (21, I guess) cycles. The 21 nucleotides are adapters and barcodes. If the reads are quality-trimmed by CLC-workbench to an average length of 288, than add this information to the Methods-section
  • Line 249: The genome properties??? Property would be DNA, size, purifiyed…. Rephrase!!! The whole sentence. I know what you mean, but you did not write it correctly
  • Line 251: Once again, I know what you mean,… The sequence breath was …. 91% were covered…. Only xxxx nucleotide were missing…. Rephrase In fact, at NCBI you wrote, that the genome is complete!!!! Now something should be missing??? Make it conclusive
  • Line 253: Which de novo assembly? Of the phage or the bacterium? Explain and provide more information. Seems, that the assembly is not correct and should be revised, or that there was a contamination of the sequencing reaction. Check in the lab, which other samples were sequenced in this run and/or with the barcode in previous runs, to proof if these contigs are real. What is the coverage? Similar to 140x? than it should be assembled in the contigs. Just a hint: Try to perform a de novo assembly with only ~60x coverage. Maybe this will make it better. You will not get less contigs, if you do not use long inserts (larger 400bp) or an additional long-read-technology. This is not necessary this time, but for the next time, it improves the sequence!!
  • Line 310 No!!! Be scientifically correct. It is not specific for B. anthracis at all. It also lysis B. cereus!! It is sensitive for B. anthracis!! This phage is interesting, but it is not highly specific for B. anthracis!

Appendix and figures:

Most of the figures are good. One can discuss, if some of them might be pushed into supplemental data. But definitively, contact somebody, who helps you a bit with plotting the graphs in R. It is just a bit, but really helpful. Or ask google. The labelling of the axis!! Look at it!! There is written e.g. “data2$Hour” No one is interested, to know that in your code you had a second dataset (“data2”) Just rename the x- and y-axis and for the legend as well

Funding:
I just saw at the Bioproject at NCBI, that the sequencing was supported by two grants. Could it be, that you missed one or both grants in the acknowledgment? Please add.

Supplemental Table:

Table S1: Reorder the agar plate-result, to get the same order as “without”-phage-experiments

Table S2:

  • Sequence reads after trim?? "Sequence reads" is correct. Trimming cuts bases of a read with low quality.
  • G/C content pXO2 is missing

Figure S2: Figure is too small. Especially phage cannot be seen, even if you zoom in. Provide high quality images like Figure S3

Author Response

Reviewer 1:

A Unique isolation of a lytic Bacteriophage Infected Bacillus anthracis Isolate from Pafuri, South Africa by Hassim et al.

The manuscript is well written and understandable to the reader. It is easy to follow the thoughts and argumentation of the authors during their presentation of the results and their discussion. In general, the presented content is worthy for publication and shows interesting ideas for future applications. It provides, insights into the bacteriophages-world of Bacillus cereus-group. However, two things are to mention prior accepting this manuscript for publication, in order to improve this study:

Alternatively: please see attachment.

Writing and Inconsistency:

  • Italize everywhere(!) Bacillus, Bacillus cereus, Bacillus anthracis etc. Good at the beginning, bad in the middle, acceptable at the end. Similar for virus-families (Myoviridae e.g.) Italize everywhere. Even in figure legends.

corrected

  • Check “comma” for “1,000”-markings. Do not know the English word of it. Some are missing. Some are present.

corrected

  • Line 82: “One gram of soil”
  • Line 90: Bacillus spp. isolated (not Isolated). Yes, MS Word has an auto-correction tool, but you have to check its suggestions.

Corrected

  • Line 134: “Nextera”

Changed to Nextera

  • Line 143: Check white space after DS201579
  • Italize “de novo” everywhere (line 149, e.g.)

Changed in the whole document

  • Line 182: Check Spelling of Laboratory. I guess you mean “Laboratory” not “laboratory”

Corrected.

  • Line 219 “indicated”

Corrected

  • Line 262 check white space before 90%

deleted

  • Line 320 Check if “.” Is subscribed or not (latter would be correct)

Scientific Remarks

  • Abstract line 19. Was this bacteriophage now environmental (soil-borne) or a veterinarian sample (wildebeest carcass). Correct the sentence.

Sentence was changed to “In this study we present the unique isolation of a dsDNA bacteriophage from a wildebeest carcass site suspected of having succumbed to anthrax.”

The wildebeest carcass was not intact, being little more than tattered skin and bone after the vulture feeding. Blood swabs are collected from within turbinate bones, orbital sockets or if the carcass is completely dry, from within the hoof where the last drops of moisture remain. In this case the swab was taken from the orbital socket of the skull.

Soil samples are taken from the clearing where it is evident the carcass first fell (also showing a ring of extravasation fluid). White backed vultures feed in large numbers and disrupt the carcass site a great deal during feeding. Generally, there is soil all over the carcass remnants. In such cases, we do not consider it a “pure” carcass sample, although we concede that calling it an environmental sample is also not strictly accurate.

  • Abstract line 26 “but more”. Just remember these two words. “more” means, that it can be counted or measured. There is no quantitative answer in this text to support this item!

Noted. The sentence has been changed to “Bacillus phage Crookii was lytic against B. cereus sensu lato group members but demonstrated a greater affinity for encapsulated B. anthracis at lower concentrations (<1x108 pfu) of virus.”

  • Line 65: “naturally occurring pathogen disinfectant agents”. Nice wording for a talk, but not for a written paper. Humans can think about it, to use it as disinfectant, but the bacteriophage does not intend to be disinfectant. It intends to survive!!

This was meant to denote the effect of the virus in the context of anthrax outbreak diagnostics and our understanding of outbreak dynamics in the environment rather than in relation to the virus by itself.

We added “….as naturally occurring pathogen disinfectant agent while the phage propagates” to indicate the phage’s intent to survive/carry out its encoded function.

  • Line 102: You tested B. cereus, 2x B. anthracis, B. thuringiensis, B. subtilis and B. mycoides. You did not test “ anthracis var cereus” That is missing definitively. Order the species to cereus-group and not-cereus group, as you mentioned, that the bacteriophage is specific to B. cereus-group members. Did you really tested B. subitilis?

We do not have access to B. anthracis var cereus and would have great difficulty bringing it into the country, even for research purposes, as our Department of Agriculture permits require BSL3 or higher for such research (related to the Animal Diseases Act legislation in our country). Because we do not have B. anthracis var cereus type isolates in our region and the downstream research component of the work is carried out in a non-endemic region. We will request a special dispensation and use your comment to motivate for future work, so many thanks for the suggestion.

Perhaps the use of B. subtilis is confusing lumped in with the rest. It was included because it is abundant on plates along the Limpopo River where the carcass was found. Also; it is used in the lab SOP’s as a control to test the efficacy of disinfection protocols on spores. Clarified in text.

  • Line 126: Nice to know that you used Rstudio, but this is not important. You used “R” in version 3.5 or similar. Mention and cite this. You even used MS Word and did not write the manuscript with Windows. Rstudio is just an environment for using R in a more easy way.

Noted, with apologies.

  • Line 135: please have a look into the recommendation of the the Nextera Kit. I am pretty sure, that you did not use 0.2ng of template!! Check this

Typing error* 1 ng DNA (5 ul) was used.

  • Line 136: “diluted in hybridization buffer” It looks, that you do not know what you did. The libraries are diluted and formamide was added, if you are using Illumina MiSeq. Including a phiX as run-control. There is not really a "hybridization" buffer. And this step belongs to “Preparing and loading of the Sequencer” not to the section “Preparing of library”. Just skip this sentence.

The sentence was removed in the manuscript.

  • Line 143: de novo assembly was performed using software suite “CLC Genomic workbench”. But which assembler was used. Please add.

CLC Genomic workbench has an inhouse built assembler which does not reveal the assembler or algorithm it uses in the tool, unlike Geneous or other purchased licences.

  • Line 143 “The assembly contigs”. Did you mean “the assembled contigs”? To be honest, I do not really know, what you did here. Did you blast the contigs in order to confirm, that it is B. anthracis? Mapping and re-ordering of contigs/scaffolds was mentioned in the next sentence

The sentence was improved to “

The assembled contigs of the bacterium were aligned with BLASTn using B. anthracis Ames Ancestor as a reference genome. MAUVE tool  was used to re-order the assembled contigs using B. anthracis Ames ancestor genome”

  • Line 146: did you really use “mauve” and not “progressive mauve”, which is implemented in mauve. Please check and correct.

The progressive mauve tool was used.

  • Line 154: “multiple alignments…was performed”. Which alignments? Replicates? Several ORFs? Which ones. Please add information.

Typo error * We performed whole genome alignment of the DS201579 and Ames ancestor using progressive mauve tool. The sentence was further clarified and the words were removed * multiple alignments*.

  • Line 157: “Bootstrap replications” not “interactions”

Changed to replication

  • Line 167: Raw reads are missing. Provide raw reads (fastq-files) to the readers by submitting them to NCBI or ENA of both samples, phage and B. anthracis

The study submitted the sequences to NCBI in a form of fasta format for the readers to access.

  • Line 175: No, this did not confirm to be B. anthracis. It preliminary identified the sample as Ba , but did not confirm it to be. No molecular results (B. anthracis specific PCR) was presented. The confirmation was done with the NGS-sequencing. Rephrase and correct the sentence. Try to be scientifically correct.

Corrected in text, with apologies.

  • Line 248: “about 140x sequenced reads”???? I know what you mean, but that is not, what you wrote. You had a coverage (sequencing depth) of 140x.

Noted, changed to coverage

  • Line 248: 280-bp??? No, you had 2x 300bp paired end reads. The sequencer sequenced 300 and something (21, I guess) cycles. The 21 nucleotides are adapters and barcodes. If the reads are quality-trimmed by CLC-workbench to an average length of 288, than add this information to the Methods-section

This information was added in the method section “The reads were quality trimmed to an average length of 288 bp”

  • Line 249: The genome properties??? Property would be DNA, size, purifiyed…. Rephrase!!! The whole sentence. I know what you mean, but you did not write it correctly

Sentence was changed to Genome features of Bacillus anthracis DS201679

  • Line 251: Once again, I know what you mean,… The sequence breath was …. 91% were covered…. Only xxxx nucleotide were missing…. Rephrase In fact, at NCBI you wrote, that the genome is complete!!!! Now something should be missing??? Make it conclusive

B. anthracis DS201579 is a draft genome sequence that consists of 31 contigs. Therefore it is not a complete genome as it reflects on NCBI (LVWF01000001-LVWF01000031). Only the pX02 was assembled as a complete plasmid (accession# CM008136).

This has been further clarified in the manuscript.

  • Line 253: Which de novo assembly? Of the phage or the bacterium? Explain and provide more information. Seems, that the assembly is not correct and should be revised, or that there was a contamination of the sequencing reaction. Check in the lab, which other samples were sequenced in this run and/or with the barcode in previous runs, to proof if these contigs are real. What is the coverage? Similar to 140x? than it should be assembled in the contigs. Just a hint: Try to perform a de novo assembly with only ~60x coverage. Maybe this will make it better. You will not get less contigs, if you do not use long inserts (larger 400bp) or an additional long-read-technology. This is not necessary this time, but for the next time, it improves the sequence!!

Line 252 was rereferring to de novo assembly of the unmapped reads achieved from read mapping analysis of the bacterium. This consisted of small contigs < 400bp a lower sequence coverage (Table S4-added in the new version of the manuscript). There were no sequence contaminants in the sequence run during indexing. The sequence data of this bacterium was well generated on MiSeq. This section is now clarified in the results section. The bacteriophages was assembled into a single contig with 70X sequence coverage. Thank you for the sage input, we will most certainly apply it in future.

  • Line 310 No!!! Be scientifically correct. It is not specific for B. anthracis at all. It also lysis B. cereus!! It is sensitive for B. anthracis!! This phage is interesting, but it is not highly specific for B. anthracis!

Agreed. Is it then fair to say? “Under such conditions, it can be said that, at low concentrations of Bacillus phage Crookii there is a higher specificity for pathogenic B. anthracis in the anthrax enzootic region of Pafuri.” which is what I initially intended it to mean.

Appendix and figures:

Most of the figures are good. One can discuss, if some of them might be pushed into supplemental data. But definitively, contact somebody, who helps you a bit with plotting the graphs in R. It is just a bit, but really helpful. Or ask google. The labelling of the axis!! Look at it!! There is written e.g. “data2$Hour” No one is interested, to know that in your code you had a second dataset (“data2”) Just rename the x- and y-axis and for the legend as well.

Corrected, with apologies.

Funding:
I just saw at the Bioproject at NCBI, that the sequencing was supported by two grants. Could it be, that you missed one or both grants in the acknowledgment? Please add.

The Bioproject (stated as using NRF and DFG funding on NCBI) was created to include other phages, from the same region, which have not been uploaded yet. The DFG grant was a grant for MLVA genotyping of Bacillus anthracis in enzootic regions as well as inducing lysogenic phages out of these strains.

This project however was carried out using the staff and resources of South African state laboratories and sequencing platforms (funded by National Research Foundation and AgriSETA) since it was during routine state surveillance and an “accidental” find. The authors (and their associated institutions) included on this submission are the only ones to contribute intellectually and financially to this paper and these findings.

The Bioproject will ultimately include the combined datasets for that region and reflect the combined funding.

Supplemental Table:

Table S1: Reorder the agar plate-result, to get the same order as “without”-phage-experiments

Table was re-ordered

Table S2:

  • Sequence reads after trim?? "Sequence reads" is correct. Trimming cuts bases of a read with low quality.
  • G/C content pXO2 is missing

GC content was added to 33

Figure S2: Figure is too small. Especially phage cannot be seen, even if you zoom in. Provide high quality images like Figure S3

Figure 2 was removed from the manuscript due to it causing confusion for the reader that the bacterium is a complete genome.

Reviewer 2 Report

Summary

The manuscript describes the isolation of a lytic anthrax phage from a carcass site in Kruger National Park. The phage is further characterized by testing it's effect on related Bacillus species and under different conditions. The phage is shown to have a higher affinity to the encapsulated strain, but also affects the unencapsulted strain as well as other closely related Bacillus spp.. Furthermore, the phage termed Crookii is analyzed by whole genome sequencing and compared to related phages.

Broad comments

Overall, I find the study interesting and providing new and relevant information to the scientific community. However, both presentation of methods and results appears unclear in places not allowing the reader to fully appreciate how the authors arrived at their conclusions (see detailed comments below). Additionally, I would recommend some form of English language review.

Specific comments

Title: Unique… Is it really unique as Bacillus phage WPh seems to have been isolated under similar conditions

Line 13: bacterium instead of bacteria (appears multiple times)

Line 89: What was the colony morphology before?

Line 101-102: How was the species of the two lab strains determined?

Line 114-126: It is not mentioned how often the experiments were repeated and how the cell counts were determined?

Line 126: Please also indicate which version of R was used.

Line 181-2: could you indicate the method of identification?

Line 219: should be "indicated"

Line 248: Was the read length really 280, normally you can obtain 300 or 250 from the described platform?

Line 249: "The genome properties…" the number of contigs is not a property of the genome, but of the quality your assembly

Line 256-257: Were the detected phages in your strain 100% identical to the given sequences from genbank? There is no indication of the match percentage in table S3. For me it is also unclear what region position refers to since you have multiple contigs in your genome?

Line 258: Here you mention unmapped reads, however in the methods section you do not mention that you performed any read mapping. Please include this in the methods section.

Line 258-263: this section seems a bit vague. How many contigs did you obtain exactly and how large were they? Could this also be a contamination?

Line 272: Please explain how the presence of tRNA genes indicates a circular genome.

Line 289ff: You say the isolation of phage Crookii was unique however you also mention that Phage W.Ph was isolated under similar conditions?

Line 310: Since you showed that it also infects B. cereus I would not see this as highly specific

Line 329: Did you confirm this difference with a statistical test?

Line 345-346: Can you rule out that phage was present outside the spores in the inoculum?

Line 354: You mention that you found a metallo-beta-lactamase. This should also be in the results section. And did you test, if it has any effect?

Line 365: It doesn't really make sense to confirm the presence of the chromosome – how would it not be present?

Line 366-367: This sentence makes no sense. Analyses cannot be found in the ancestor…

Table S1 and AppendixA: To what volume do the cell counts refer (cells per ?) and how were they determined?

AppendixA: Labelling and description of the plots is very confusing. Why do you sometimes show log and ln transformed data? Did you do any statistical testing?

Figure S2: It is very difficult to see anything in this figure.

Figure S3: It is not indicated how figure S3 was created

Author Response

Reviewer 2

Alternatively: please see attachment

Summary

The manuscript describes the isolation of a lytic anthrax phage from a carcass site in Kruger National Park. The phage is further characterized by testing it's effect on related Bacillus species and under different conditions. The phage is shown to have a higher affinity to the encapsulated strain, but also affects the unencapsulted strain as well as other closely related Bacillus spp.. Furthermore, the phage termed Crookii is analyzed by whole genome sequencing and compared to related phages.

Broad comments

Overall, I find the study interesting and providing new and relevant information to the scientific community. However, both presentation of methods and results appears unclear in places not allowing the reader to fully appreciate how the authors arrived at their conclusions (see detailed comments below). Additionally, I would recommend some form of English language review.

Specific comments

Title: Unique… Is it really unique as Bacillus phage WPh seems to have been isolated under similar conditions.

The similar conditions are that they were identified from (1) swabs from carcass sites (2) during an active anthrax outbreak (3) in enzootic regions of National Parks in southern Africa. Although Dr Beyer’s research group noticed the activity of the phage on bacterial culture, there is no report (verbal or written) where such a high level of bacteriophage activity was observed from samples at the carcass site…within soil samples or on the swab itself.

Beyer et al, 2012 is a genotyping and epidemiology paper of B. anthracis strains in Namibia from which Phage WPh was apparently isolated from. There is no mention of the phage in the paper.

Klumpp et al, 2014 compares the characterization of 3 tailed phages (of which WPh is 1).

Jakub Barylski and Annika Gillis’ publications are about distribution, characterisation and classification of bacteriophages that infect Bacillus spp.

This isolation is unique in the over 40 year diagnostic history of anthrax surveillance in Kruger National Park. There may have been others that were missed in the past…which is what makes this one noteworthy. There is also no such record in any publications that we could find which describes the implications of such bacteriophage activity in complicating diagnostic results in a natural outbreak in the field. Nor for the overall implications for anthrax outbreak/transmission dynamics in enzootic regions.

Line 13: bacterium instead of bacteria (appears multiple times)

corrected

Line 89: What was the colony morphology before?

The colonies were initially typical in morphology, but thereafter took on a shinier or more turbid appearance on the blood agar…I believe this was due to bacterial lysis being underway because plaques would become apparent after some time (as seen in Figure 2).

Line 101-102: How was the species of the two lab strains determined?

It was determined at our Bacteriology diagnostic unit through classic bacteriologic techniques (now indicated in the text) i.e. gram stain, motility, colony morphology, media utilization/selection and/or API kits if necessary etc.

Line 114-126: It is not mentioned how often the experiments were repeated and how the cell counts were determined?

“Blood smears (5 slides for each set) were made with 20 µL of blood at 8 hours, then 12 hours and at 24 hours. These smears were immediately fixed in methanol and stained with Romanowsky-Giemsa for 30 minutes [32] and examined at 1000x magnification. Microscopic fields were manually counted on each slide.”

Line 126: Please also indicate which version of R was used.

Added in text

Line 181-2: could you indicate the method of identification?

It was determined at our Bacteriology diagnostic unit through classic bacteriologic techniques (now indicated in the text) i.e. gram stain, motility, colony morphology, media utilization/selection and/or API kits if necessary etc

Line 219: should be "indicated"

Corrected

Line 248: Was the read length really 280, normally you can obtain 300 or 250 from the described platform?

This information was added in the method section “The reads were quality trimmed to an average length of 288 bp”

Line 249: "The genome properties…" the number of contigs is not a property of the genome, but of the quality your assembly

The sentence is improved and now reads as “A 140X sequenced coverage of  the  trimmed 280-bp paired end reads were used to assemble the 31 contigs of B. anthracis DS201579 genome”

Line 256-257: Were the detected phages in your strain 100% identical to the given sequences from genbank? There is no indication of the match percentage in table S3. For me it is also unclear what region position refers to since you have multiple contigs in your genome?

The  prophages were not 100% identical to the given sequence from GenBank phages. The prophages regions of DS201579 strain are now updated using PHASTER tool which furthermore indicates the match percentage and the region positions of prophages.

Line 258: Here you mention unmapped reads, however in the methods section you do not mention that you performed any read mapping. Please include this in the methods section.

It has been included as “Sequence read mapping analysis of the DS201579 reads was performed on CLC Genomic Workbench 7.5.1 using B. anthracis Ames Ancestor (GenBank accession: NC_007530.2; NC_007322.2 and NC_007323.2) as a reference” Furthermore, unmapped reads were collected to determine the possibility of remnants of phage infection.

Line 258-263: this section seems a bit vague. How many contigs did you obtain exactly and how large were they? Could this also be a contamination?

A total of 8 contigs were determined in the unmapped reads with less than 400 bp. The assembled 8 contigs had a BLASTn homolog to partial/minor phages similar to Phage WPh. Table S4 was included in the corrected manuscript, which included the assembled contigs and BLASTn results.

Line 272: Please explain how the presence of tRNA genes indicates a circular genome.

tRNA genes doesn’t indicate circular genome *Typo error*, the statement was removed.

Line 289ff: You say the isolation of phage Crookii was unique however you also mention that Phage W.Ph was isolated under similar conditions?

As indicated above; the manner in which this phage was identified is what makes it unique. We have never before recorded such difficulty in obtaining a bacterial isolate from a carcass site, especially from soil and swabs. Had it been in the non-enzootic region or outside of an active outbreak, it probably would have been missed. None of the staff in Kruger National Park have observed such a site before, but probably due to oversight. The anthrax surveillance system has been in practice since the late 1950’s in the Park.

The similarity is only in that they are both anthrax sites of ungulates from enzootic regions of southern African National Parks where swabs produced the B. anthracis isolate as well as the phage. The disease dynamics in the 2 Parks as well as the SNP lineages of the B. anthracis hosts in Etosha and Kruger are vastly different which makes the sequence similarity (even at 85% sequence identity) of the viruses surprising.

Line 310: Since you showed that it also infects B. cereus I would not see this as highly specific

Agreed. Is it then fair to say? “Under such conditions, it can be said, that at low concentrations of Bacillus phage Crookii there is a higher specificity for pathogenic B. anthracis in the anthrax enzootic region of Pafuri.” which is what I initially intended it to mean.

Line 329: Did you confirm this difference with a statistical test?

Unfortunately, in my zeal to test environmental conditions, I did not have enough spores to do sufficient replicates to obtain a sample size for statistically significant results. In our laboratories, we cannot work with large spore numbers of pathogenic strains as this is prohibited within our permits. I therefore used descriptive statistics to make sense of vegetative cell and spore counts on fixed blood slides. And observations of phage plaques (zone diameter) on bacterial lawns.

I have corrected the sentence to “Bacillus phage Crookii proved more lytic than Gamma phage to DS201579 based on plaque zones at the same bacteriophage concentration.” in the text, but can remove it if this is not acceptable.

Line 345-346: Can you rule out that phage was present outside the spores in the inoculum?

Pre-heat treatment at either 65°C or 72°C ruled out the presence of viable phage outside of the spore at the time of inoculation since the virus denatures at temperatures higher than 60°C.

Line 354: You mention that you found a metallo-beta-lactamase. This should also be in the results section. And did you test if it has any effect?

These will be tested and characterised further in the future, but have not been recorded at this time.

Line 365: It doesn't really make sense to confirm the presence of the chromosome – how would it not be present?

Bacillus anthracis DS201579 genome features consists of the chromosome, pX01 and pX02.

Line 366-367: This sentence makes no sense. Analyses cannot be found in the ancestor…

Rectified to “Comparative genomic analyses of the prophages determined in this study were also reported present in B. anthracis Ames ancestor”

Table S1 and AppendixA: To what volume do the cell counts refer (cells per ?) and how were they determined?

Clarified to cells per 100uL of blood:

“Blood smears (5 slides for each set) were made with 20 µL of blood at 8 hours, then 12 hours and at 24 hours. These smears were immediately fixed in methanol and stained with Romanowsky-Giemsa for 30 minutes [32] and examined at 1000x magnification. A 100 microscopic fields were directly counted across the slides for each set of conditions.”

AppendixA: Labelling and description of the plots is very confusing. Why do you sometimes show log and ln transformed data? Did you do any statistical testing?

The log data was to normalise the distribution and have been re-done

Figure S2: It is very difficult to see anything in this figure.

Figure 2 was removed from the manuscript due to it causing confusion for the reader that the bacterium is a complete genome.

Figure S3: It is not indicated how figure S3 was created.

Indicated in text

Round 2

Reviewer 2 Report

I am thanking the authors for considering my suggestions. In my opinion the manuscript is much improved. There remain some minor points.

Line 149ff: This sentence appears wrong grammatically. You probably mean to say: the quality of the reads was assessed?
Table S2 mentions 572 coding sequences in the genome, in the text it is 5724 (which makes more sense)
Appendix A still appears highly confusing. There may also have been a problem with the pdf creation as some of the plots are only halfway visible and it is difficult to tell which of them were now removed. Also some labels remain confusing such as dataDS$hour (I am aware this is what R writes by default but it can be changed)

Author Response

Thank you for your suggestions and the errors you picked up. It is much appreciated.

Line 149ff: This sentence appears wrong grammatically. You probably mean to say: the quality of the reads was assessed?

Yes, thank you. Corrected in text (now line 142)

Table S2 mentions 572 coding sequences in the genome, in the text it is 5724 (which makes more sense)

It is 5724.

Appendix A still appears highly confusing. There may also have been a problem with the pdf creation as some of the plots are only halfway visible and it is difficult to tell which of them were now removed. Also some labels remain confusing such as dataDS$hour (I am aware this is what R writes by default but it can be changed

My apologies, I left the track changes in for review and the pdf document has thus included the old figures. I have now accepted the track changes from the last round and saved it to reflect the figures for inclusion which had been re-labelled. I hope this is acceptable.